

# Association between the CASC16 rs4784227 polymorphism and breast cancer risk and prognosis in a northeast Chinese Han population

Yue Zhang[1,2], Changgui Kou[2], Lin Jia[1], Yangyang Gao[1], Xin Li[2], Hao Wu[2], Naifei Chen[1] and Zheng Lv[1]

[1] Cancer Center, The First Affiliated Hospital of Jilin University, Changchun, China
[2] Department of Epidemiology and Biostatistics, School of Public Health, Jilin University, Changchun, China

## ABSTRACT

**Background.** Breast cancer (BC) poses a serious threat to women worldwide. This research was designed to explore the association between the rs4784227 polymorphism of cancer susceptibility candidate gene 16 (*CASC16*) and BC susceptibility and prognosis, aiming to provide further information for the early detection of BC and to accelerate comprehensive cancer management.

**Methods.** A total of 1,733 subjects were recruited for this case-control study, of which 828 are BC patients and 905 are healthy individuals. The relevance between SNP rs4784227 and BC risk in diverse genetic models was analyzed by using the SNPStats analysis program and was assessed by odds ratios (ORs) and 95% confidence intervals (CIs) using the binary logistic regression model. Pearson's $\chi^2$ test was used to determine the correlation between the polymorphism and clinical characteristics of BC patients. Additionally, univariate survival analysis was performed by the Kaplan-Meier method and log-rank test, and multivariate survival analysis was performed by Cox regression.

**Results.** SNP rs4784227 was significantly associated with susceptibility to BC in the dominant model (CT/TT versus CC, OR = 1.237, 95% CI = 1.012–1.513, P = 0.038). The minor allele of SNP rs4784227 was significantly linked to an increased risk of BC (OR = 1.197, 95% CI = 1.022–1.401, P = 0.026). In addition, the rs4784227 polymorphism of *CASC16* was associated with perineural invasion (P = 0.030), menstrual status (P = 0.016) and histological grade (P = 0.001, P = 0.003, P = 0.025; respectively) of BC patients. There was no significant association between the genotypes of rs4784227 and disease-free survival (DFS) or overall survival (OS) of breast cancer patients (P > 0.05).

**Conclusions.** The rs4784227 polymorphism of *CASC16* may affect susceptibility to breast cancer and is associated with perineural invasion, menstrual status and histological grade in BC patients. Additionally, our results could not confirm that this polymorphism was related to breast cancer prognosis.

Corresponding author
Zheng Lv, lvz@jlu.edu.cn

## INTRODUCTION

Breast cancer (BC) is the most common cause of cancer death in women and has become the most common form of cancer worldwide. Similarly, the estimated age-standardized incidence rate of BC is the highest among all cancers in China (*Sung et al., 2021*). The recurrence rate of patients with early breast cancer after surgery is relatively low (*Weigel & Dowsett, 2010*). According to information from the 2019 Chinese Society of Clinical Oncology (CSCO) Breast Cancer Annual Meeting, the 5-year survival rate of breast cancer in China was more than 80%. The increase in the cure rate of breast cancer in China depends on early detection, early diagnosis and early treatment. To improve the BC survival rate, early screening and early detection are critical and are designed primarily to identify those at high risk.

High-risk groups can be identified by risk factors such as female sex, early menarche, late menopause, increased estrogen exposure and excessive alcohol intake (*Washbrook, 2006*; *Horn & Vatten, 2017*; *Dall & Britt, 2017*; *Jung et al., 2016*). Additionally, gene detection is also feasible. Previous studies have shown that people who carry the breast cancer susceptibility gene 1 (*BRCA1*) or breast cancer susceptibility gene 2 (*BRCA2*) variants are predisposed to breast cancer (*Kurian et al., 2014*). This is an indication that single nucleotide polymorphism (SNP) studies could help identify people with high susceptibility to breast cancer.

Cancer susceptibility candidate gene 16 (*CASC16*), is one of these genes that have been studied. *CASC16* is an RNA gene that is located at chromosome 16q12.1 and is affiliated with the lncRNA class. One study showed that the *CASC16* gene had higher expression in breast cancer cells than in normal cells (*Han et al., 2016*), and some loci of *CASC16* have been demonstrated to be significantly associated with BC susceptibility (*Long et al., 2010*; *Zuo et al., 2020*; *Tajbakhsh et al., 2019*; *Lin et al., 2014*). However, the association between this gene and other cancers has not yet been found. Rs4784227 is a locus of the *CASC16* gene and previous studies showed that its MAF in Asian groups was in a range from 0.20 to 0.28 (https://www.ncbi.nlm.nih.gov/snp/rs4784227). It is well known that *FOXA1* is involved in *ESR1*-mediated transcription, the regulation of cell apoptosis and cell cycle regulation. The place for rs4784227 on *FOXA1* genomic for interaction is on the eighth position of the *FKH* motif recognized via *FOXA1*, thus, affinity DNA site for *FOXA* protein was enhanced for the T allele compared with the C allele in rs4784227, which suggested that the locus may affect a DNA binding sequence change on *FOXA1* and modulate the chromatin affinity for *FOXA1* (*Tajbakhsh et al., 2019*; *Cowper-Sal Lari et al., 2012*). However, whether the interaction between *FOXA1* and *CASC16* participates in the above functions and what will be caused by it are not clear. Current studies are limited to the relationship between this locus and BC susceptibility, and the functions of *CASC16* are still unknown (*Zuo et al., 2020*).

Moreover, rs4784227 has been known to be linked to BC risk in previous studies conducted in Chinese populations, such as Shanghai, Tianjin, Nanjing and Taiwan (OR>1) (*Long et al., 2010*). However, these subject groups are mainly located in southern and coastal cities, and relatively little research has been done on inland areas. Investigations

of this locus using a northeast Chinese population may verify whether the findings previously identified are generalizable to other populations. Data from a study published previously indicated that the rs4784227 SNP in *CASC16* promotes lymph node metastasis in BC patients (*Sun et al., 2020*). In addition, stratified analysis using a genetic model indicated that rs4784227 was specific to progesterone receptor positivity (*Deng et al., 2016*). In addition, there have been no studies of the correlation between this locus and BC survival. Thus, in this article, we further explored the clinical features of SNP rs4784227 and its relationship with BC prognosis.

## SUBJECTS AND METHODS

### Study population

In total, 828 BC patients confirmed by pathological cytology and histology and 905 healthy individuals from the First Affiliated Hospital of Jilin University (Changchun, Jilin Province, China), who were all genetically unrelated Han Chinese and originated from the Northeast, were recruited in this case-control study. The sample size was calculated by Quanto and the power of our study was above 90%. In addition, the clinical characteristics of patients with BC, including age, menopausal status, family history, pathological type, histological grade, tumor size, lymph node metastasis status, lymph vascular space invasion and perineural invasion, were collected through medical records, which were registered between April 2013 and September 2016. Among these clinical characteristics, breast cancer staging relies on the TNM system based on the sixth edition of AJCC and histological grade was based on Nottingham grading system.

### Selection of SNP and genotyping

Rs4784227 was chosen based on a published paper that reported that this SNP might be related to BC susceptibility. Genomic DNA was extracted from peripheral blood samples from all the study participants. Moreover, we used the MassAray system (Agena, San Diego, CA, United States) as well as the matrix-assisted laser desorption ionization-time of flight (MALDI-TOF) mass spectrometry method to detect genotypes of the rs4784227 locus of *CASC16*. SNP genotyping was conducted without knowing cases and controls' status.

### Ethical approval

Written informed consent was obtained from all participants of this study, and this study was approved by the Institutional Ethics Committee of The First Affiliated Hospital of Jilin University with the ethical approval number 2014-031. All the methods were carried out in accordance with the guidelines of Helsinki's declaration.

### Statistical analysis

Data analyses were conducted using SPSS 24.0 software (IBM Corp, Armonk, NY, USA) and the online platform SNPStats (http://www.snpstats.net/start.htm). Hardy-Weinberg equilibrium (HWE) was examined by the chi-squared test in both cases and controls. The optimum model of the locus was determined according to the AIC (Akaike's information criterion) value. The genotype and allele frequency distribution of the rs4784227 genetic

polymorphism in the case group and control group in different genetic models was analyzed by the SNPStats analysis program, and corresponding odds ratios (ORs) and 95% confidence intervals (CIs) adjusted by age were computed using the binary logistic regression model. In addition, Pearson's $\chi^2$ test was used to compare the frequency distributions of various clinical characteristics of the different genotypes of rs4784227.

In addition, survival analysis was used to further determine the impacts of SNP rs4784227 on BC prognosis by selecting disease-free survival (DFS) and overall survival (OS) as prognostic indicators. The time of the surgery was the start of DFS, and the data from follow-ups were retrospectively collected. Univariate survival analysis was performed by the Kaplan–Meier method and log-rank test, and survival curves were drawn by R software (*R Core Team, 2022*). Additionally, multivariate survival analysis was performed by Cox's proportional hazards regression model. All statistical tests were two-tailed, and $P < 0.05$ was considered statistically significant.

## RESULTS

### Clinical characteristics of participants

The median age of the case group was 51 (44,58) and that of the control group was 38 (32,53). The clinical characteristics of BC cases are presented in Table 1.

### Analysis of the association between the rs4784227 polymorphism of *CASC16* and BC risk

The genotype distribution of cases and controls suggested that SNP rs4784227 conformed to the HWE ($P > 0.05$). The association between SNP rs4784227 and BC susceptibility in different genetic models is shown in Table 2. The results suggested that the rs4784227 polymorphism of *CASC16* may be associated with BC risk. After adjusting for age, the results of binary logistic regression analysis showed that the CT/TT genotype of rs4784227 significantly increased susceptibility to BC compared with the CC genotype ($OR_{CT/TTvsCC} = 1.237$, 95% CI $= 1.012$–$1.513$, $P = 0.038$). The T allele was the minor allele of rs4784227, which was significantly associated with an increased risk of BC compared with the C allele (OR $= 1.197$, 95% CI $= 1.022$–$1.401$, $P = 0.026$). Furthermore, the dominant model was the best-fitting model of rs4784227 (AIC $= 2190.4$).

### The association between the rs4784227 polymorphism of *CASC16* and clinical characteristics of breast cancer

The results of some of these characteristics that were statistically significant or studied previously are shown in Table 3. The findings showed that BC patients who carried the CC genotype were more at risk for perineural invasion than those who carried the CT/TT genotype ($P = 0.030$). Moreover, the proportion of premenopausal people who carried the CT genotype was smaller than that in the CC and TT genotypes and was larger than that in the CC and TT genotypes in postmenopausal people ($P = 0.016$). In addition, the proportion of people who carried the C allele was smaller than that of those carrying the T allele with a histological grade I tumor and was larger than that of those carrying the T allele with a histological grade III tumor ($P = 0.001$). The proportion of people who

**Table 1  Clinical characteristics of breast cancer patients.**

| Characteristics | Case, n (%) |
|---|---|
| Age (years) | |
| ≤35 | 52 (6.28) |
| >35 | 776 (93.72) |
| Menopausal status | |
| Premenopause | 398 (48.07) |
| Postmenopause | 430 (51.93) |
| Family history | |
| No | 796 (96.14) |
| Yes | 32 (3.86) |
| Molecular type | |
| LuminalA | 104 (12.56) |
| LuminalB | 520 (62.80) |
| HER2 | 103 (12.44) |
| Triple negative | 101 (12.20) |
| Pathological type | |
| Infiltrating ductal carcinoma | 793 (95.77) |
| Other types | 35 (4.23) |
| Histological grade | |
| I | 31 (3.74) |
| II | 511 (61.72) |
| III | 286 (34.54) |
| Tumor size | |
| T1 | 422 (50.97) |
| T2 | 365 (44.08) |
| T3 | 27 (3.26) |
| T4 | 14 (1.69) |
| Lymph node metastasis | |
| No | 396 (47.83) |
| Yes | 432 (52.17) |
| Lymph-vascular space invasion | |
| No | 471 (56.88) |
| Yes | 357 (43.12) |
| Perineural invasion | |
| No | 689 (83.21) |
| Yes | 139 (16.79) |
| Treatment protocol | |
| Only anthracycline | 30 (3.63) |
| Only taxane | 131 (15.84) |
| Anthracycline and taxane | 462 (55.86) |
| Others | 204 (24.67) |
| Endocrinetherapy | |
| Aromatase inhibitor | 218 (26.36) |
| Tamoxifen | 199 (24.06) |
| Aromatase inhibitor and tamoxifen | 14 (1.69) |
| No | 396 (47.89) |

**Table 2 The relationship between the rs4784227 polymorphism of *CASC16* and the risk of breast cancer.**

| Allele/Genotype | | Case, n (%) | Control, n (%) | OR (95%CI) | P | AIC |
|---|---|---|---|---|---|---|
| Allele | C | 1158 (69.93) | 1336 (73.81) | 1.000 | 0.026 | – |
| | T | 498 (30.07) | 474 (26.19) | 1.197 (1.022–1.401) | | |
| Genotype | | | | | | |
| Codominant | CC | 401 (48.43) | 493 (54.48) | 1.000 | 0.105 | 2191.7 |
| | CT | 356 (43.00) | 350 (38.67) | 1.221 (0.989–1.507) | 0.063 | |
| | TT | 71 (8.57) | 62 (6.85) | 1.328 (0.904–1.952) | 0.148 | |
| Dominant | CC | 401 (48.43) | 493 (54.48) | 1.000 | 0.038 | 2190.4 |
| | CT/TT | 427 (51.57) | 412 (45.52) | 1.237 (1.012–1.513) | | |
| Recessive | CC/CT | 757 (91.43) | 843 (93.15) | 1.000 | 0.306 | 2192.6 |
| | TT | 71 (8.57) | 62 (6.85) | 1.215 (0.837–1.765) | | |
| Overdominant | CC/TT | 472 (57.00) | 555 (61.33) | 1.000 | 0.121 | 2193 |
| | CT | 356 (43.00) | 350 (38.67) | 1.175 (0.958–1.441) | | |

**Notes.**

Note: *ORs* and *P* values of the genotypes were adjusted by age.

carried the CC genotype was smaller than that in those carrying the TT genotype with a histological grade I tumor and was larger than that in those carrying the TT genotype with a histological grade III tumor ($P = 0.003$). Furthermore, the number of people who carried the CC genotype was less than those carrying the CT/TT genotype with a histological grade I tumor and was more than those carrying the CT/TT genotype with a histological grade III tumor ($P = 0.025$).

## The relationship between the rs4784227 polymorphism of *CASC16* and the prognosis of breast cancer

Among the 828 BC patients, 401 individuals were carrying the CC genotype, and the others were carriers of the CT or TT genotype in the dominant model. By using the log-rank test, we did not find that DFS and OS in BC cases were significantly associated with the rs4784227 polymorphism of *CASC16* (DFS: $P = 0.972$, OS: $P = 0.727$) (Fig. 1).

All the above clinical characteristics of patients were included in the Cox model for multivariate analysis to examine the association between the genotypes of rs4784227 and DFS and OS of BC. The results suggested that the differences in DFS and OS among the different genotypes of the BC patients were not statistically significant (DFS: $HR = 1.037$ (0.705–1.526) $P = 0.855$, OS: $HR = 1.224$ (0.653–2.295) $P = 0.529$).

## DISCUSSION

To date, genome-wide association studies (GWAS) have identified numerous common variants associated with BC risk at multiple genetic loci (*Deng et al., 2016*). Therefore, conducting research on SNPs could help identify people with a high susceptibility to BC. As mentioned before, rs4784227 may affect the DNA binding sequence on *FOXA1* and subsequently increase the *FOXA1*-binding affinity to the *CASC16* gene promoter; *FOXA1* plays an important role in the function of ER and growth of ER$^+$ BC cells (*Carroll et al., 2005*; *Kong et al., 2011*). BC patients with ER and/or PR positivity accounted for

**Table 3  Correlations between SNP rs4784227 and the clinical characteristics of patients with breast cancer.**

| Characteristics | | | Allele | | Genotype | | | Genotype | |
|---|---|---|---|---|---|---|---|---|---|
| | | | C | T | CC | CT | TT | CC | CT/TT |
| ER | Status | Negative | 312 (26.94) | 116 (23.29) | 113 (28.18) | 86 (24.16) | 15 (21.13) | 113 (28.18) | 101 (23.65) |
| | | Positive | 846 (73.06) | 382 (76.71) | 288 (71.82) | 270 (75.84) | 56 (78.87) | 288 (71.82) | 326 (76.35) |
| | $\chi^2$ | | 2.420 | | 2.494 | | | 2.210 | |
| | $P$ | | 0.120 | | 0.287 | | | 0.137 | |
| PR | Status | Negative | 474 (40.93) | 184 (36.95) | 168 (41.90) | 138 (38.76) | 23 (32.39) | 168 (41.90) | 161 (37.70) |
| | | Positive | 684 (59.07) | 314 (63.05) | 233 (58.10) | 218 (61.24) | 48 (67.61) | 233 (58.10) | 266 (62.30) |
| | $\chi^2$ | | 2.309 | | 2.519 | | | 1.516 | |
| | $P$ | | 0.129 | | 0.284 | | | 0.218 | |
| HER2 | Status | Negative | 787 (67.96) | 339 (68.07) | 272 (67.83) | 243 (68.26) | 48 (67.61) | 272 (67.83) | 291 (68.15) |
| | | Positive | 371 (32.04) | 159 (31.93) | 129 (32.17) | 113 (31.74) | 23 (32.39) | 129 (32.17) | 136 (31.85) |
| | $\chi^2$ | | 0.002 | | 0.021 | | | 0.010 | |
| | $P$ | | 0.965 | | 0.989 | | | 0.922 | |
| Lymph node metastasis | Status | No | 553 (47.75) | 239 (47.99) | 191 (47.63) | 171 (48.03) | 34 (47.89) | 191 (47.63) | 205 (48.01) |
| | | Yes | 605 (52.25) | 259 (52.01) | 210 (52.37) | 185 (51.97) | 37 (52.11) | 210 (52.37) | 222 (51.99) |
| | $\chi^2$ | | 0.008 | | 0.012 | | | 0.012 | |
| | $P$ | | 0.929 | | 0.994 | | | 0.913 | |
| Family history | Status | No | 1120 (96.72) | 472 (94.78) | 389 (97.01) | 342 (96.07) | 65 (91.55) | 389 (97.01) | 407 (95.32) |
| | | Yes | 38 (3.28) | 26 (5.22) | 12 (2.99) | 14 (3.93) | 6 (8.45) | 12 (2.99) | 20 (4.68) |
| | $\chi^2$ | | 3.525 | | 4.845 | | | 1.592 | |
| | $P$ | | 0.060 | | 0.089 | | | 0.207 | |
| Perineural invasion | Status | No | 951 (82.12) | 427 (85.74) | 322 (80.30) | 307 (86.24) | 60 (84.51) | 322 (80.30) | 367 (85.95) |
| | | Yes | 207 (17.88) | 71 (14.26) | 79 (19.70) | 49 (13.76) | 11 (15.49) | 79 (19.70) | 60 (14.05) |
| | $\chi^2$ | | 3.264 | | 4.851 | | | 4.724 | |
| | $P$ | | 0.071 | | 0.088 | | | 0.030 | |
| Tumor size | Status | T1 | 590 (50.95) | 254 (51.00) | 202 (50.37) | 186 (52.25) | 34 (47.89) | 202 (50.37) | 220 (51.52) |
| | | ≥T2 | 568 (49.05) | 244 (49.00) | 199 (49.63) | 170 (47.75) | 37 (52.11) | 199 (49.63) | 207 (48.48) |
| | $\chi^2$ | | <0.001 | | 0.559 | | | 0.109 | |
| | $P$ | | 0.984 | | 0.756 | | | 0.741 | |
| Menstrual status | Status | Premenopause | 559 (48.27) | 237 (47.59) | 203 (50.62) | 153 (42.98) | 42 (59.15) | 203 (50.62) | 195 (45.67) |
| | | Postmenopause | 599 (51.73) | 261 (52.41) | 198 (49.38) | 203 (57.02) | 29 (40.85) | 198 (49.38) | 232 (54.33) |
| | $\chi^2$ | | 0.065 | | 8.241 | | | 2.035 | |
| | $p$ | | 0.799 | | 0.016 | | | 0.154 | |

**Table 3** (*continued*)

| Characteristics | | | Allele | | Genotype | | | Genotype | |
|---|---|---|---|---|---|---|---|---|---|
| | | | C | T | CC | CT | TT | CC | CT/TT |
| Histological grade | Status | I | 31 (2.68) | 31 (6.22) | 8 (2.00) | 15 (4.21) | 8 (11.27) | 8 (2.00) | 23 (5.39) |
| | | II | 714 (61.66) | 308 (61.85) | 246 (61.35) | 222 (62.36) | 43 (60.56) | 246 (61.35) | 265 (62.06) |
| | | III | 413 (35.66) | 159 (31.93) | 147 (36.65) | 119 (33.43) | 20 (28.17) | 147 (36.65) | 139 (32.55) |
| | $\chi^2$ | | 13.118 | | 15.751 | | | 7.379 | |
| | $P$ | | 0.001 | | 0.003 | | | 0.025 | |
| Molecular type | Status | LuminalA | 138 (11.92) | 70 (14.06) | 44 (10.97) | 50 (14.04) | 10 (14.08) | 44 (10.97) | 60 (14.05) |
| | | LuminalB | 723 (62.44) | 317 (63.65) | 250 (62.35) | 223 (62.64) | 47 (66.20) | 250 (62.34) | 270 (63.23) |
| | | HER2 | 141 (12.18) | 65 (13.05) | 47 (11.72) | 47 (13.21) | 9 (12.68) | 47 (11.73) | 56 (13.12) |
| | | Triple negative | 156 (13.46) | 46 (9.24) | 60 (14.96) | 36 (10.11) | 5 (7.04) | 60 (14.96) | 41 (9.60) |
| | $\chi^2$ | | 6.685 | | 7.372 | | | 6.782 | |
| | $P$ | | 0.083 | | 0.288 | | | 0.079 | |

**Notes.**

ER, estrogenic receptor; PR, progesterone receptor; HER2, human epidermal growth factor receptor 2.

approximately 75% of BC patients (*Niemeier et al., 2010*), and ER is an important indicator of treatment efficacy prediction and prognosis. Therefore, we wanted to know whether there was a connection between the rs4784227 polymorphism of *CASC16* and susceptibility to BC, especially ER$^+$BC. Thus, we selected the GWAS-identified SNP rs4784227 to investigate and verify its association with BC susceptibility and prognosis in the northeast Chinese Han population.

In our study, we found that the T allele was the minor allele of rs4784227, and the distribution of alleles and genotypes of SNP rs4784227 ($C = 69.93\%$, $T = 30.07\%$, CC $= 48.43\%$, CT $= 43.00\%$, TT $= 8.57\%$) was consistent with current research. The results indicated that those who carried the T allele had high BC susceptibility, which was in line with *Zuo et al. (2020)*, *Tajbakhsh et al. (2019)* and *He et al. (2014)*. In our study, the best-fitting inheritance model of rs4784227 was the dominant model, and in this model, we found that the CC genotype of SNP rs4784227 provided a protective effect against BC, which was consistent with *Zuo et al. (2020)* and *Tajbakhsh et al. (2019)*. However, *Sun et al. (2020)* did not find this association. The difference may be because the sample size in Sun's research was relatively small ($n_{Zuo} = 681$, $n_{Tajbakhsh} = 505$, $n_{Sun} = 503$), so the results of the data analysis may be biased. The results of the northwest population in our study were the same as the populations in the southern and coastal cities in China, Japan and Europe. Therefore, racial and regional differences may not affect the association between this locus and BC susceptibility.

Regarding clinical features, *He et al.*'s (*2014*) study ($n = 623$) found that the T allele of rs4784227 exhibited significant associations with the status of ER, PR and HER2 in an additive model. In addition, *Deng et al. (2016)* indicated that SNP rs4784227 had a significant association with PR-positive tumor risk. Again, rs4784227 may affect the interactions between *FOXA1* and *CASC16*, and *FOXA1* may affect the status of ER. Thus,

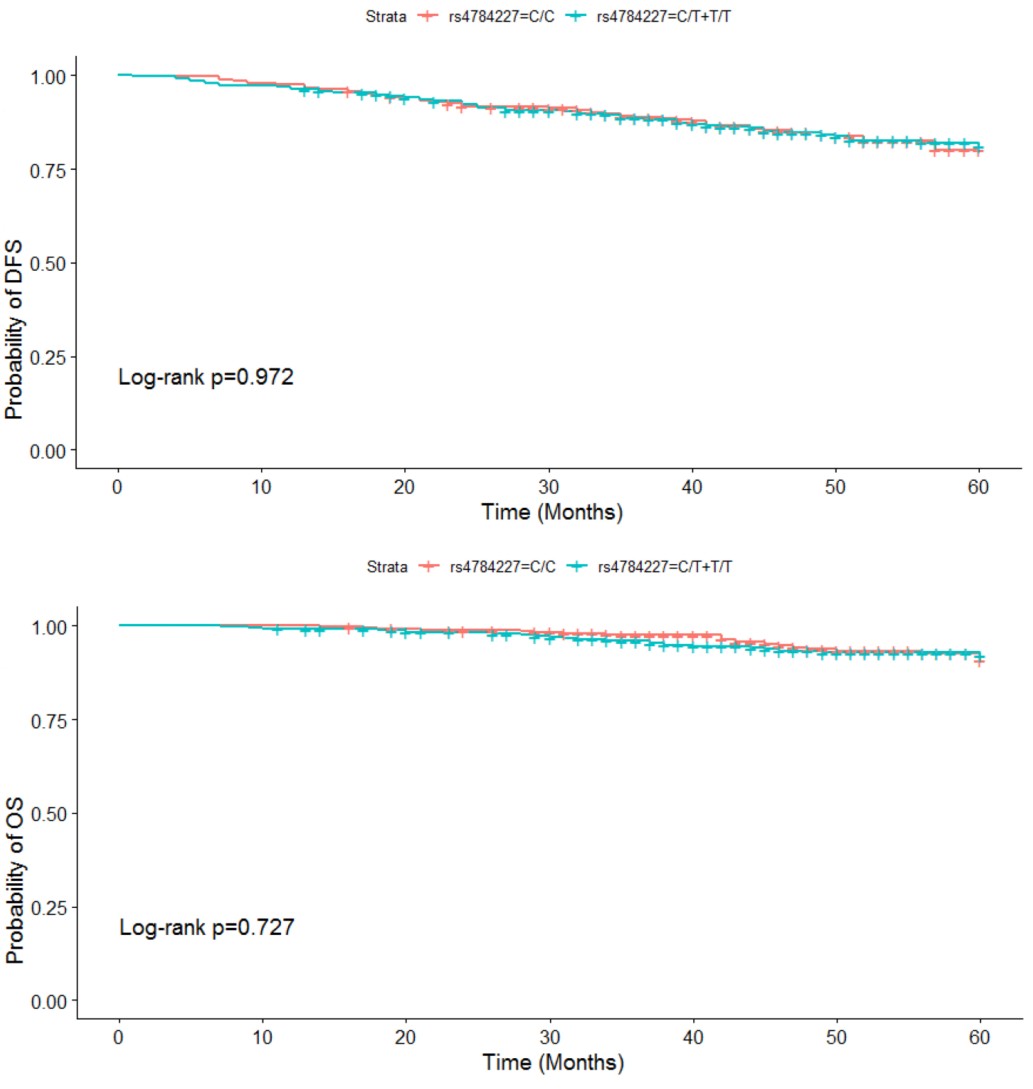

**Figure 1** **DFS and OS for patients with CC, CT and TT genotypes of the rs4784227 polymorphism of *CASC16*.**

we investigated whether SNP rs4784227 could affect the expression of these three receptors, especially ER. Regrettably, in our study, this association could not be found in any of the three receptors. *He et al.*'s *(2014)* research suggested that rs4784227 increased BC risk in a dose-dependent manner, so homozygous carriers have higher susceptibility. Therefore, the difference mentioned above may be because the number of people who carried the TT genotype in our sample was small (8.57%), which was not enough to prove the relationship between this locus and the status of ER. Therefore, studies on larger sample sets and further functional identification of molecular mechanisms are needed to confirm the association mentioned above. Furthermore, the results indicated that the locus was associated with perineural invasion, menstrual status and histological grade of BC patients. Among them, the menstrual status results were consistent with *Lin et al. (2014)*. However, the same

findings of perineural invasion and histological grade were not found in other studies. In addition, *Zuo et al. (2020)* and *Sun et al. (2020)* found that there was a significant association between rs4784227 and lymph node metastasis status, but this conclusion was not reached in our study. We propose that this may be because the number of people who carried the TT genotype in our sample was small. Therefore, larger sample sizes and a meta-analysis are required to further verify this relationship.

Additionally, our study could not confirm that the rs4784227 polymorphism of *CASC16* had an effect on the DFS and OS of BC patients. This may be because the sample time span was just 60 months and the disease-free survival of BC was long, so a longer follow-up period may be necessary.

## CONCLUSIONS

Our findings suggest that the rs4784227 polymorphism of *CASC16* may affect susceptibility to BC and was associated with perineural invasion, menstrual status and histological grade of BC patients. In addition, our results could not confirm that the rs4784227 polymorphism of *CASC16* had an effect on the prognosis of breast cancer.

### Funding

This work was supported by Jilin Provincial Department of science and technology [grant numbers 20200201474JC]. The funders had no role in study design, data collection and analysis, decision to publish, or preparation of the manuscript.

### Grant Disclosures

The following grant information was disclosed by the authors:
Jilin Provincial Department of science and technology: 20200201474JC.

### Competing Interests

The authors declare there are no competing interests.

### Author Contributions

- Yue Zhang performed the experiments, analyzed the data, prepared figures and/or tables, authored or reviewed drafts of the article, and approved the final draft.
- Changgui Kou conceived and designed the experiments, authored or reviewed drafts of the article, and approved the final draft.
- Lin Jia analyzed the data, prepared figures and/or tables, authored or reviewed drafts of the article, and approved the final draft.
- Yangyang Gao performed the experiments, authored or reviewed drafts of the article, and approved the final draft.
- Xin Li performed the experiments, prepared figures and/or tables, and approved the final draft.

- Hao Wu analyzed the data, authored or reviewed drafts of the article, and approved the final draft.
- Naifei Chen conceived and designed the experiments, prepared figures and/or tables, and approved the final draft.
- Zheng Lv conceived and designed the experiments, authored or reviewed drafts of the article, and approved the final draft.

## Human Ethics

The following information was supplied relating to ethical approvals (i.e., approving body and any reference numbers):

This study was approved by the Institutional Ethics Committee of The First Affiliated Hospital of Jilin University.

## Data Availability

The raw data was available in the Supplementary File.

## Supplemental Information

Supplemental information for this article can be found online at http://dx.doi.org/10.7717/peerj.14462#supplemental-information.

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
