# Peer review of "Association between the CASC16 rs4784227 polymorphism and breast cancer risk and prognosis in a northeast Chinese Han population"

_PeerJ, doi:10.7717/peerj.14462_

## Round 0.1 · original submission · Major Revisions

Please address comments 1,3 and 4 in 'Validity of the findings' from Reviewer 1 and all comments in 'Experimental Design' from Reviewer 2.

Reviewer 2 has asked for additional context for validity of the findings, including demographic and clinical information as well as hormonal factor information. Please try to provide as much detail on the study as you can with regard to these.

Further information on sample size calculation is essential as a supplementary note.

Reviewer 1 ·

Basic reporting

Overall, this manuscript is clearly written in professional. Language is concise and easy to follow. The authors used GWAS to investigate the risk association between SNP rs4784227 and breast cancer. However, there are some weaknesses that the authors need to revise,

Experimental design

the experimental design is generally good with sufficient case numbers.

Validity of the findings

1.Please make a table to summarize the characteristics of the 828 breast cancer patients. For example, how many patients are in luminal A, luminal B, Her2+, and TNBC cancer? How many patients are BRCA1/2 mutation carriers?
2. Lines 57-58, have never heard that people who carry CDH1 gene are susceptible to breast cancer ? Normally, CDH1 (E-Cadherin) high expression promotes cell-cell tight junction which limits cancer cell migration and metastasis.
3. Among the 828 breast cancer patients, please show the association between SNP rs4784227 and luminal A, luminal B, HER2+, and TNBC.
4. For the survival curve, please show the correlation between SNP rs4784227 and luminal A, luminal B, HER2+, and TNBC.

·

Basic reporting

Introduction: 1. The introduction part requires more focused and depth towards the pleiotropy of the two genes and their impact in general. It will be good if these two genes mutation synergistically found in other diseases /mechanism/Populations (CASC16 and FOXA1).

2. The study has targeted single SNP, so it will be good, if provides detailed background regarding the SNP. When we talk in terms of SNP, it would be good to provide MAF population-wise as well as morbidities reporting the same SNP.

3. English is good throughout, but spell checking and grammar checking recommended throughout the text.

Experimental design

1. It is important to provide sample size calculation and power of study detailed in main text as well as in supplementary material.

2. Please provide the detail of methods of confirmation of breast cancer among patients.

3. Please explain in main text, what clinical classification used for TNM, staging, grading of the cancer.

4. Please also discuss clinical information in the main text in methodology in 3.1 clinical characteristics of the patients, provide detail like % molecular subtypes of breast cancer, TNM, Staging, grading, gender, hormonal status etc.

5. Also describe demographic features of the present population and also perform statical analysis. As it is well established that confounding factors also influence genetic susceptibility.

6.

Validity of the findings

Population based study requires detailed information regarding the clinical as well as demographic factors associated with SNPs. Besides this hormonal factor should also be included.

Provide detailed description in text regarding the medication therapy neo-adjuvant therapy or multimodal therapy.

The author should check statistically all above-mentioned factors and also advised to check if there is multi-modal treatment related or hormonal therapy related influence on the survival of the patients.

Additional comments

After including all these factors. It is highly recommended t provide detailed description of these in all the sections of the main and supplementary text wherever applicable.

---

## Round 0.2 · accepted · Accept

Thank you for addressing all of the reviewers' comments. I have assessed the revision myself and the previous concerns about validity of the findings as well as the experimental design- along with power calculations and demographics/clinical information have been thoroughly addressed and enriches the quality of your manuscript.

I believe that your findings can enable further functional studies of rs4784227 in the CASC16 locus in breast cancer models to advance therapeutics and biomarker discovery. The manuscript is ready for publication in its current state.

Reviewer 1 ·

Basic reporting

The authors answered my questions. I recommend to accept this manuscript.

Experimental design

The authors answered my questions. I recommend to accept this manuscript.

Validity of the findings

The authors answered my questions. I recommend to accept this manuscript.

Additional comments

The authors answered my questions. I recommend to accept this manuscript.